# Absence of central tolerance in Aire-deficient mice synergizes with immune-checkpoint inhibition to enhance antitumor responses

Asiel A. Benitez[1], Sara Khalil-Agüero[1], Anjali Nandakumar[1], Namita T. Gupta[1], Wen Zhang[1], Gurinder S. Atwal[1], Andrew J. Murphy [1], Matthew A. Sleeman[1] & Sokol Haxhinasto [1✉]

The endogenous anti-tumor responses are limited in part by the absence of tumor-reactive T cells, an inevitable consequence of thymic central tolerance mechanisms ensuring prevention of autoimmunity. Here we show that tumor rejection induced by immune checkpoint blockade is significantly enhanced in *Aire*-deficient mice, the epitome of central tolerance breakdown. The observed synergy in tumor rejection extended to different tumor models, was accompanied by increased numbers of activated T cells expressing high levels of Gzma, Gzmb, Perforin, Cxcr3, and increased intratumoural levels of Cxcl9 and Cxcl10 compared to wild-type mice. Consistent with Aire's central role in T cell repertoire selection, single cell TCR sequencing unveiled expansion of several clones with high tumor reactivity. The data suggest that breakdown in central tolerance synergizes with immune checkpoint blockade in enhancing anti-tumor immunity and may serve as a model to unmask novel anti-tumor therapies including anti-tumor TCRs, normally purged during central tolerance.

---

[1] Regeneron Pharmaceuticals, Inc. 777 Old Saw Mill River Road, Tarrytown, NY 10591, USA. ✉email: sokol.haxhinasto@regeneron.com

The immune system has evolved in response to the threat of pathogenic microorganisms by carefully regulating these responses to clear the pathogen but also to prevent immunity from damaging the host. The latter is primarily achieved by establishing immune tolerance, a two-pronged endeavor consisting of central and peripheral tolerance, ensuring that "self-reactive" lymphocytes are either eliminated or subjected to regulatory programs to limit any negative impact to the host. Defects in the pathways involved in the establishment and maintenance of tolerance give rise to autoimmunity, often being associated with damage to specific tissues as a result of infiltrating autoreactive lymphocytes.

In addition to protecting the body from pathogens, it is now recognized that the immune system also plays a role in cancer surveillance by removing cancerous, damaged, or stressed cells that through modifications or mutations present molecules or antigens that are deemed as nonself[1]. However, cancerous cells often escape detection and clearance by the immune system through mechanisms involving immune editing, enhanced regulatory T cell activity, and increased surface expression or secretion of suppressive molecules[2]. Advances made over the last two decades in the field of oncology have greatly enhanced the survival rates of cancer patients. These advances include immunotherapies such as antibodies blocking the cytotoxic T lymphocyte-associated protein 4 (CTLA-4) and the programmed cell death 1 (PD-1), and the development of chimeric antigen receptor-modified T cells[3,4]. Despite these advances, cancer remains the second leading cause of deaths in the United States and therefore represents a tremendous burden on society[5]. The high incidence of cancer and limited responses to current therapies necessitates the development of new therapies to improve patient survival.

Autoimmune regulator (Aire) plays a key role in establishing central T cell tolerance thereby protecting against the development of autoimmunity. Mutations in AIRE cause autoimmunity against different organs in humans and mice[6–8] accompanied by a spectrum of manifestations, such as, autoimmune polyglandular syndrome type-1 (APS-1), hypothyroidism, Addison's disease, and others[9]. Recently it was also shown that pneumonitis is prevalent in patients with APS-1 characterized by high levels of activated neutrophils and lymphocytes[10]. Aire is expressed in medullary thymic epithelial cells (mTECs) where it functions as a transcriptional regulator promoting the expression of tissue-restricted self-antigens (TSAs). Self-reactive thymocytes that recognize these TSAs with high affinity are eliminated through apoptosis or differentiate into regulatory T cells (Tregs)[11]. Several reports have also shown that both in mice and humans, Aire is also expressed in secondary lymphoid organs by a specialized population of cells, namely eTACs (extra-thymic Aire+ cells), with a suggested role in regulating tolerance[12–15], albeit this contribution remains an open question.

Prior studies have indicated that deficiency in Aire promotes the clearance of melanomas, due to the presence of self-reactive T cells capable of recognizing self-antigens expressed on melanoma cells[16–18]. In addition, in vivo depletion of mTECs expressing Aire using anti-RANKL antibodies resulted in enhanced clearance of melanoma cells[19]. Furthermore, in humans, single-nucleotide polymorphisms in AIRE have been shown to be protective against melanoma[20]. Here we demonstrate that breakdown in central tolerance in Aire-deficient mice, results in enhanced rejection of multiple tumor types importantly in the presence of different immune-checkpoint inhibitors. In an effort to delineate the cellular and molecular mechanisms responsible for this synergy we performed single-cell RNAseq and TCRseq. We show that the antitumor response in Aire−/− mice is greatly augmented, composed of multiple pro-inflammatory cell types enlisting multiple pathways with the potential to serve as future combination therapies.

## Results

**Aire deficiency results in potent antitumor rejection in combination with PD-1 blockade.** To evaluate whether defects in central tolerance in combination with immune-checkpoint inhibition affected tumor growth, Aire+/+ or Aire−/− mice were implanted with the poorly immunogenic MC38 colon tumor model[21,22]. PD-1 is a receptor expressed on T cells that is induced upon activation resulting in decreased proliferation, cytotoxicity, and cytokine production[23]. Mice were injected with anti-PD1 or Isotype (Rat IgG2a) antibodies at the indicated time points and tumor growth kinetics were monitored (Fig. 1a). Consistent with prior reports[24], we observed that PD1 blockade had a modest reduction in tumor growth in Aire+/+ mice (Fig. 1b and Supplementary Fig. 1a), whilst this difference was greatly augmented in Aire−/− mice. Analysis of the tumor infiltrates revealed that tumors from wild-type animals treated with anti-PD1 consisted of significantly more infiltrating CD8+ T cells as previously shown[21] (Fig. 1c). However, Aire−/− mice treated with anti-PD1 had a significantly higher percentage of CD8+ T cells (15% vs. 10%), and an increase in the CD8/CD4 ratio and CD8/Treg ratio compared with wild-type mice treated with anti-PD1 (Fig. 1c, e, and Supplementary Fig. 1c). No major differences were observed in the CD4+ TIL population (Fig. 1d). Importantly, the observed increase was restricted to the tumors, as we did not observe any marked differences in the levels of splenic CD8+ or CD4+ T cells suggesting the response is driven by specific tumor antigens in the tumors from Aire−/− (Supplementary Fig. 1d–f).

To further profile the observed antitumor response, we performed RNAseq in tumors from Aire+/+ and Aire−/− mice treated with anti-PD1. This showed that tumors from Aire-deficient mice had a larger number of upregulated genes compared with the wild-type mice (Supplementary Fig. 2a, b). There was an enrichment of genes involved in T cell activity (Fig. 2a) as well as chemokines and chemokine receptors (Fig. 2b). There was a significant increase in the number of genes associated with antitumor immune responses such as Cd3e, Cd8α, Ifnγ, Tnfα, and FasL (Fig. 2c). Furthermore, there was an increase in the expression of chemokines such as Xcl1, Cxcl9, Cxcl10, and chemokine receptor Cxcr3 in the tumors from Aire−/− (Supplementary Fig. 2c). High levels of Cxcl9 and Cxcl10 in tumors correlate with increased recruitment of CD8+ T cells expressing Cxcr3[25,26]. Interestingly, chemokine profiling revealed higher levels of CXCL10 in the serum from Aire-/- mice (Supplementary Fig. 2d) in agreement with previous reports showing high levels of CXCL10 in APS-1 patients[27], suggesting a potential mechanisms for the enhanced antitumor response. In addition, tumors from Aire−/− mice had lower levels of expression of Ptp4a1 and Meis2 which have been shown to promote tumor progression and are associated with poor survival[28,29] (Supplementary Fig. 2e).

**Aire deficiency results in potent melanoma rejection in combination with immune-checkpoint blockade.** We next wanted to test whether Aire−/− mice also displayed increased antitumor activity against B16F10 melanoma. To this end, we implanted mice with B16.F10 cells and treated mice with anti-CTLA4 or isotype antibodies on days 3, 7, 10, and 14 and tumor growth kinetics were monitored. Consistent with published results[16], anti-CTLA4 treatment had a profound effect on tumor growth in the Aire−/− mice compared with wild type (Fig. 3a). Interestingly, Aire−/− mice treated with isotype displayed increased antitumor activity over the wild type also treated with isotype control

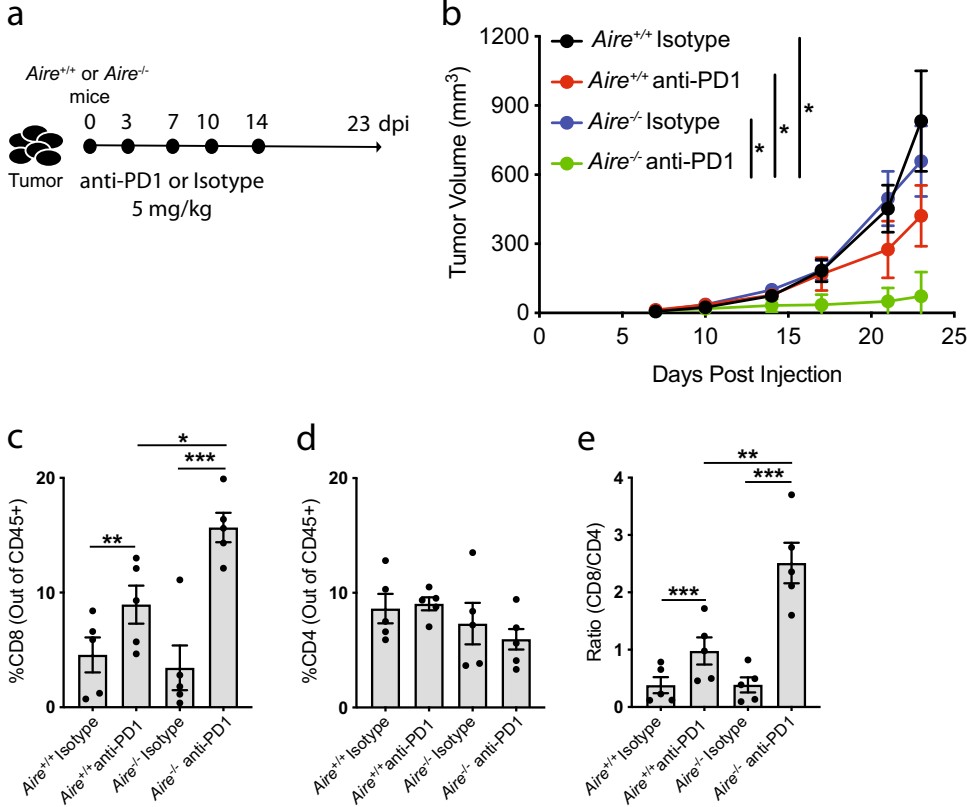

**Fig. 1 Aire⁻/⁻ mice displayed increased tumor killing in combination with PD-1 blockade. a** Schematic depicting antibody treatment regimen in *Aire*⁺/⁺ and *Aire*⁻/⁻ implanted with MC38. Mice were injected with Isotype or anti-PD1 antibodies at 5 mg/kg on days 0, 3, 7, 10, and 14. **b** Growth kinetics of MC38 tumors in *Aire*⁺/⁺ and *Aire*⁻/⁻ treated with Isotype or anti-PD1 ($n = 8$ per group). **c, d** Percentage of intra-tumoral CD8⁺ or CD4⁺ T cells from *Aire*⁺/⁺ and *Aire*⁻/⁻ treated with isotype or anti-PD1 ($n = 5$ per group) at day 23. Cells were gated on live cells and percentages were determined out of total CD45⁺ cells. **e** Ratio of CD8⁺ to CD4⁺ T cells. Data are represented as mean ± SEM, (*$P < 0.05$; **$P < 0.01$; ***$P < 0.001$), one-way ANOVA with Tukey's test.

antibody. Profiling of tumor infiltrates revealed that anti-CTLA4 blockade increased the percentage of both CD4⁺ and CD8⁺ T cells in wild-type mice over isotype (Supplementary Fig. 3a, b). However, *Aire*⁻/⁻ mice treated with anti-CTLA4 had a significantly higher percentage of CD4⁺ and CD8⁺ T cells and an increase in the CD8/CD4 ratio and CD8/Treg ratio over the wild type also treated with anti-CTLA4 (Supplementary Fig. 3c, d). We observed similar results using the MC38 tumor model in *Aire*⁺/⁺ and *Aire*⁻/⁻ mice treated with anti-CTLA4 or isotype antibodies (Supplementary Fig. 3e). We next wanted to determine if the enhanced antitumor activity of *Aire*⁻/⁻ mice could be unmasked with another immune-checkpoint inhibitor. To this end, we implanted mice with B16F10 melanoma tumors treated with anti-PD1 or isotype antibodies on days 3, 7, 10, and 14 and tumor growth kinetics were monitored. This line of experimentation showed that *Aire*⁻/⁻ mice displayed reduced tumor growth in the presence of anti-PD1 antibody over the wild type (Fig. 3b). We next wanted to determine if there were differences in the levels of cytotoxicity of CD8⁺ TILs from *Aire*⁺/⁺ and *Aire*⁻/⁻ mice. To this end, we isolated CD8⁺ TILs from *Aire*⁺/⁺ and *Aire*⁻/⁻ mice treated with anti-PD1 or isotype antibodies, mixed with tumor cells and measured the activity of lactate dehydrogenase (LDH), which is released from the cytosol of damaged cells into the supernatant[30]. This line of experimentation showed that direct cytotoxicity testing against B16F10 target cells resulted in enhanced activity for CD8⁺ TILs from *Aire*⁻/⁻ mice treated with anti-PD1 as compared with wild type when effector-to-target ratios were at least 7.5:1 (Fig. 3c). In addition, there was an

increase in the levels of IFNγ and TNFα released by the CD8⁺ TILs from *Aire*⁻/⁻ mice treated with anti-PD1 compared with wild type (Supplementary Fig. 3f, g). Interestingly, the levels of the two cytokines analyzed were similar between CD8⁺ TILs from *Aire*⁻/⁻ mice treated with anti-PD1 or isotype suggesting that the cytolytic activity of CD8⁺ TILs *Aire*⁻/⁻ mice is greatly enhanced by antagonizing the PD1 pathway.

To further elucidate the magnitude of the anti-melanoma response observed in *Aire*⁻/⁻ mice, we performed RNAseq in tumors from *Aire*⁺/⁺ and *Aire*⁻/⁻ mice treated with anti-CTLA4. These results revealed that tumors from *Aire*-deficient mice had a larger number of upregulated genes compared with the wild-type mice (Fig. 4a and Supplementary Fig. 4a). Similar to our prior observations with the MC38 model, there was a significant increase in the number of genes associated with antitumor immune responses such as Ifnγ, Gzmb, Perforin, Cxcl9, Cxcl10, and levels of Cxcr3 trended higher in *Aire*⁻/⁻ mice (Fig. 4b), supporting the hypothesis that defects in central tolerance provide a conducive immunological milieu that synergizes with various immune-checkpoint therapies across different tumor types.

**Single-cell analysis reveals drastic changes in tumor infiltrates from Aire⁻/⁻ mice.** To further elucidate the cellular and molecular changes that may account for the enhanced antitumor activity observed in *Aire*⁻/⁻ mice, we performed single-cell RNAseq on infiltrates from MC38 tumors. *Aire*⁺/⁺ and *Aire*⁻/⁻

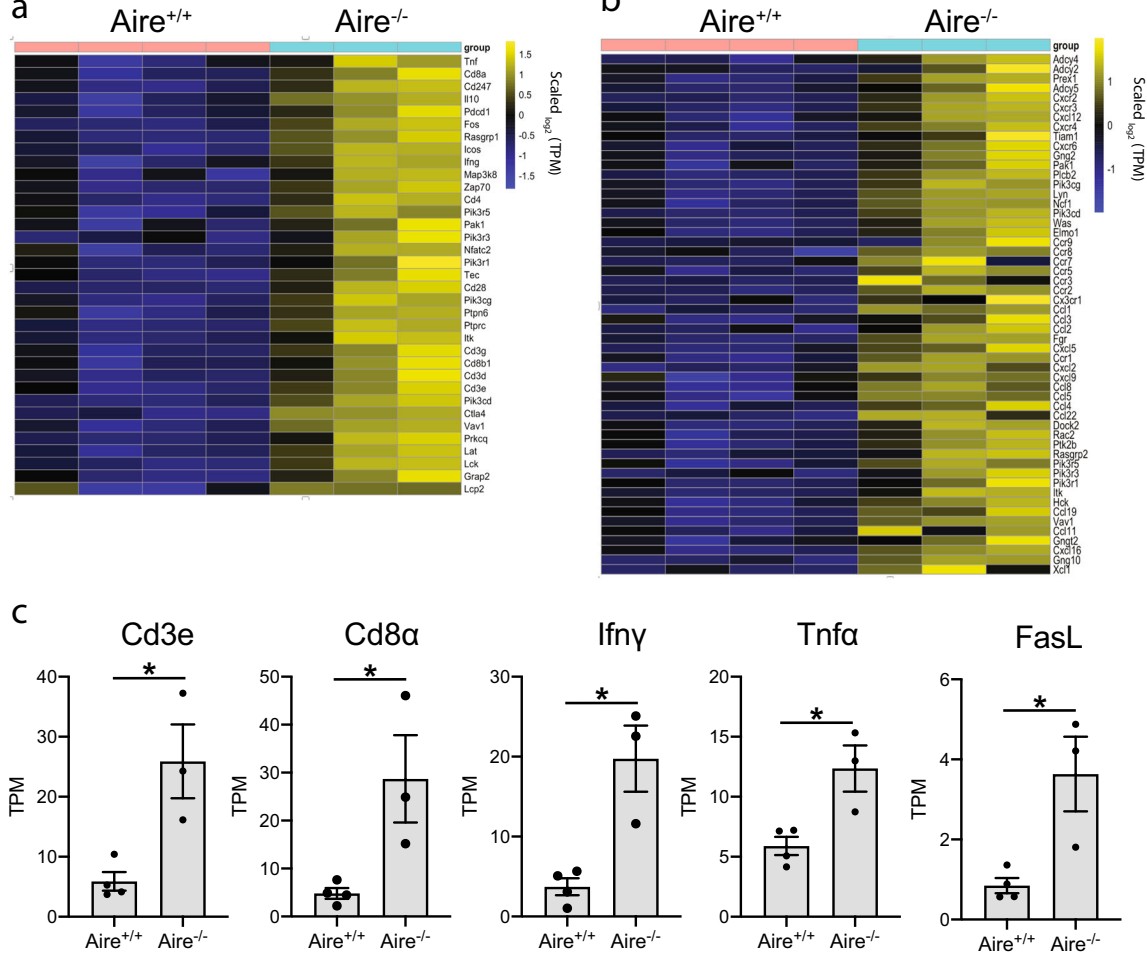

**Fig. 2 Tumors from _Aire_⁻/⁻ mice have increased levels of cytotoxic genes. a, b** Heatmaps depicting differentially regulated genes associated with T cell receptor signaling or chemokine signaling in tumors from _Aire_+/+ ($n = 4$) and _Aire_⁻/⁻ ($n = 3$) mice treated with anti-PD1. Differentially regulated genes were defined with log2 fold changes of 1.5 in either up or down direction and with $p$ values <0.01; TPM transcripts per million. The expression value of each gene was divided by the median expression of the same gene across all samples. **c** Transcript levels of Cd3e, Cd8α, Ifnγ, Tnfα, and FasL in tumors from _Aire_+/+ and _Aire_⁻/⁻ mice. TPM transcripts per million. Data are represented as mean ± SEM, (*$P < 0.05$; **$P < 0.01$; ***$P < 0.001$), by Student's $t$ test. TPM values are provided in Supplementary Data 1.

mice were implanted with tumors and anti-PD1 or isotype antibodies administered on days 0, 3, 7, 10, and 14 and tumor infiltrates were analyzed on day 19. We sorted the intra-tumoral and splenic CD45+ populations (Supplementary Fig. 5a) and individually sequenced cells from each tissue (Supplementary Table 1). Unbiased clustering was performed using the Seurat package with cell populations visualized in UMAP dimensionality reduction plots. The analysis revealed ten distinct clusters in the combined intra-tumoral and splenic CD45+ populations, each with a unique gene expression profile (Fig. 5a). The clustering of spleens and tumors separately revealed the source of the clusters identified (Supplementary Fig. 5b). Major cell types comprised T cells (CD3e+), B cells (CD79a+), macrophage/myeloid (Msr1+ C1qb+), dendritic cells (Clec9a+), pDCs (Siglech+), NK cells (Ncr1+), and neutrophils (S100a8+) (Fig. 5b). Based on specific cell markers, T cells were furthered clustered into activated CD8+ (Clusters 0 and 2), CD8+ naïve (Cluster 6), CD4+ T cells (Cluster 10), and regulatory T cells. The clusters identified expressed markers associated with each specific cell type such as Cd4, Cd8α, and Foxp3.

Clustering of each individual condition tested revealed drastic changes in percentages of several populations in _Aire_⁻/⁻ mice

(Fig. 5c). We observed that anti-PD1 treatment resulted in increased levels of activated CD8+ T cells in both _Aire_+/+ and _Aire_⁻/⁻ compared with the isotype treated animals (Clusters 0 and 2). However, there was a ~3 and ~3.6-fold increase in the percentage of activated CD8+ T cells in tumors from _Aire_⁻/⁻ mice treated with anti-PD1 over wild type (Fig. 5d). Congruent with the initial observations made in Fig. 1, we did not observe any changes in the populations identified in the spleen suggesting the response is tumor specific (Supplementary Fig. 5c).

The activated CD8+ T cells (Cluster 0) in tumors from _Aire_⁻/⁻ mice treated with anti-PD1 had higher levels of the antitumor genes Gzmb, and Perforin compared to wild type (Supplementary Table 2). In addition, the activated CD8+ T cells from Cluster 2 had increased levels of Gzma as well as Stat1 (Supplementary Table 3), the latter which has been shown to be necessary for the development of antitumor cytolytic activity[31]. Furthermore, these activated CD8+ T cells expressed high levels of the integrin Cd103, the transcription factor Egr1, and reduced levels of Klf2 which have been associated with the formation of tumor-resident memory cells ($T_{RM}$)[32]. $T_{RM}$ cells have recently been shown to be retained within the tumor microenvironment and their presence is associated with better survival[32,33].

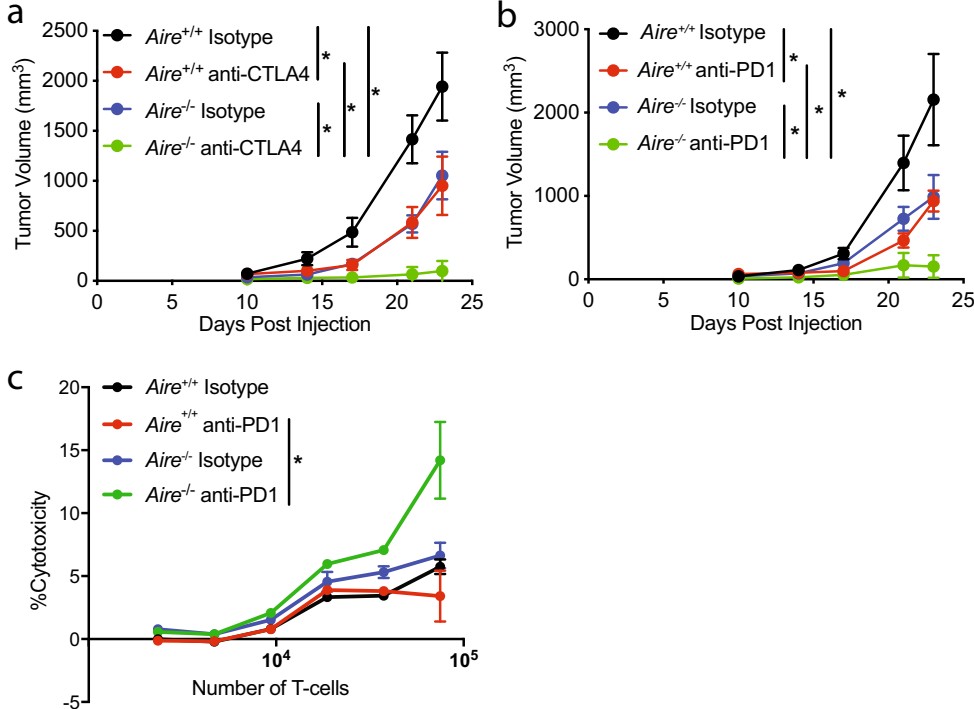

**Fig. 3 Aire⁻/⁻ mice displayed increased melanoma killing in combination with PD-1 or CTLA4 blockade. a** Growth kinetics of B16.F10 tumors in *Aire*⁺/⁺ and *Aire*⁻/⁻ treated with Isotype or anti-CTLA4 (*n* = 9 per group). **b** Growth kinetics of B16.F10 tumors in *Aire*⁺/⁺ and *Aire*⁻/⁻ treated with Isotype or anti-PD1 (*n* = 10 per group). **c** Increased cytotoxicity of CD8⁺ TILs from *Aire*⁻/⁻ mice treated with anti-PD1 (*n* = 3 per group). Data are represented as mean ± SEM, (*$P < 0.05$; **$P < 0.01$; ***$P < 0.001$), by one-way ANOVA with Tukey's test.

There were also changes in the levels of NK cells, with the *Aire*⁻/⁻ mice treated with anti-PD1 having more than the wild type (Fig. 5c, d). These NK cells express high levels of the cell-cycle regulator Pim-1 (Supplementary Table 4), and its over-expression has been suggested to protect NK cells from apoptosis[34,35]. Intra-tumoral macrophages have been shown to constitute the largest population and undergo drastic changes following immune-checkpoint therapy[36]. Our results showed that anti-PD1 treatment reduced the number of macrophages in tumors from both *Aire*⁺/⁺ and *Aire*⁻/⁻ compared with the isotype treated groups. However, the number of macrophages was strikingly lower with ~3.8-fold less in the tumors from *Aire*⁻/⁻ mice treated with anti-PD1 over the wild type (Fig. 5d). This decrease was also confirmed by determining the percentage of tumor infiltrating macrophages (Supplementary Fig. 5d). Despite the lower numbers, the macrophages present had increased levels of interferon stimulated genes such as Ifitm1, Gpb2, Gbp4, and Ifi47 (Supplementary Data 3). In addition, there were increased levels of the M1 macrophage marker Nos2 and high expression of the T cell chemoattractant Cxcl9. M1 macrophages produce high amounts of ROS and nitrogen radicals which have strong tumoricidal activity, and the secretion of Cxcl9 recruits Th1 cells to the tumor[37,38]. Altogether, these data suggest that the combination of immune-checkpoint blockade and breakdown in central tolerance orchestrates a multicellular recruitment of different immune cell types with pro-inflammatory characteristics at the tumor site and allows for unmasking of potential novel approaches for combination therapies, beyond T cells.

**Expansion of unique CD8⁺ TILs in *Aire*⁻/⁻ mice.** Given the remarkable changes in the levels of CD8⁺ T cells in the tumors from *Aire*⁻/⁻ treated with anti-PD1 compared with wild type, we wanted to determine if there were qualitative changes in sub-populations that may account for the results seen. To this end, we sorted, captured, and sequenced intra-tumoral or splenic CD8⁺ T cells from *Aire*⁺/⁺ and *Aire*⁻/⁻ treated with anti-PD1 or iso-type resulting in individually sequenced cells from each tissue respectively (Supplementary Table 5). We performed clustering of the intra-tumoral and splenic CD8⁺ T cells using the Seurat package which identified 14 unique cell clusters (Fig. 6a). The major separation of clusters was primarily driven by the tissue source of the CD8⁺ T cells, and all conditions tested were well represented in both tissues (Supplementary Fig. 6a, b). The intra-tumoral CD8⁺ T cells expressed very low levels of the naïve T cell marker *L-selectin* (CD62L) and high levels of the activation marker Cd69 (Fig. 6b). In addition, CD8⁺ TILs were character-ized as having a distinctly pro-inflammatory signature with high levels of 4-1BB, Lag3, Gzma, Gzmb, Gzmk, and Gzmc (Fig. 6b). We next determined the composition of each condition per cluster to identify any changes. We found that more than 50% of cluster 6 was composed of CD8⁺ TILs from *Aire*⁻/⁻ treated with anti-PD1 (Fig. 6c and Supplementary Fig. 6c). These cells were characterized as expressing high levels of genes associated with T cell activation such as Lmna, Gzma, Nr4a1, Cd69, Pdcd1, Tnf, and FasL (Supplementary Data 4). Furthermore, these cells had increased levels of the chemokine receptors Cxcr3 and Cxcr6. Although not as striking as cluster 6, we also observed CD8⁺ TILs from *Aire*⁻/⁻ mice treated with anti-PD1 were overrepresented in cluster 9. Interestingly, these cells expressed high levels of inter-feron stimulated genes such as Cxcl10, Isg15, Ifit3, Ifit1, Rsad2, and Mx1 (Supplementary Data 5). Type-I interferons have been shown to act directly on CD8⁺ T cells where they promote activation and differentiation[39].

Given the central role of Aire in shaping the TCR repertoire to remove autoreactive T cells, we calculated the TCR alpha and beta

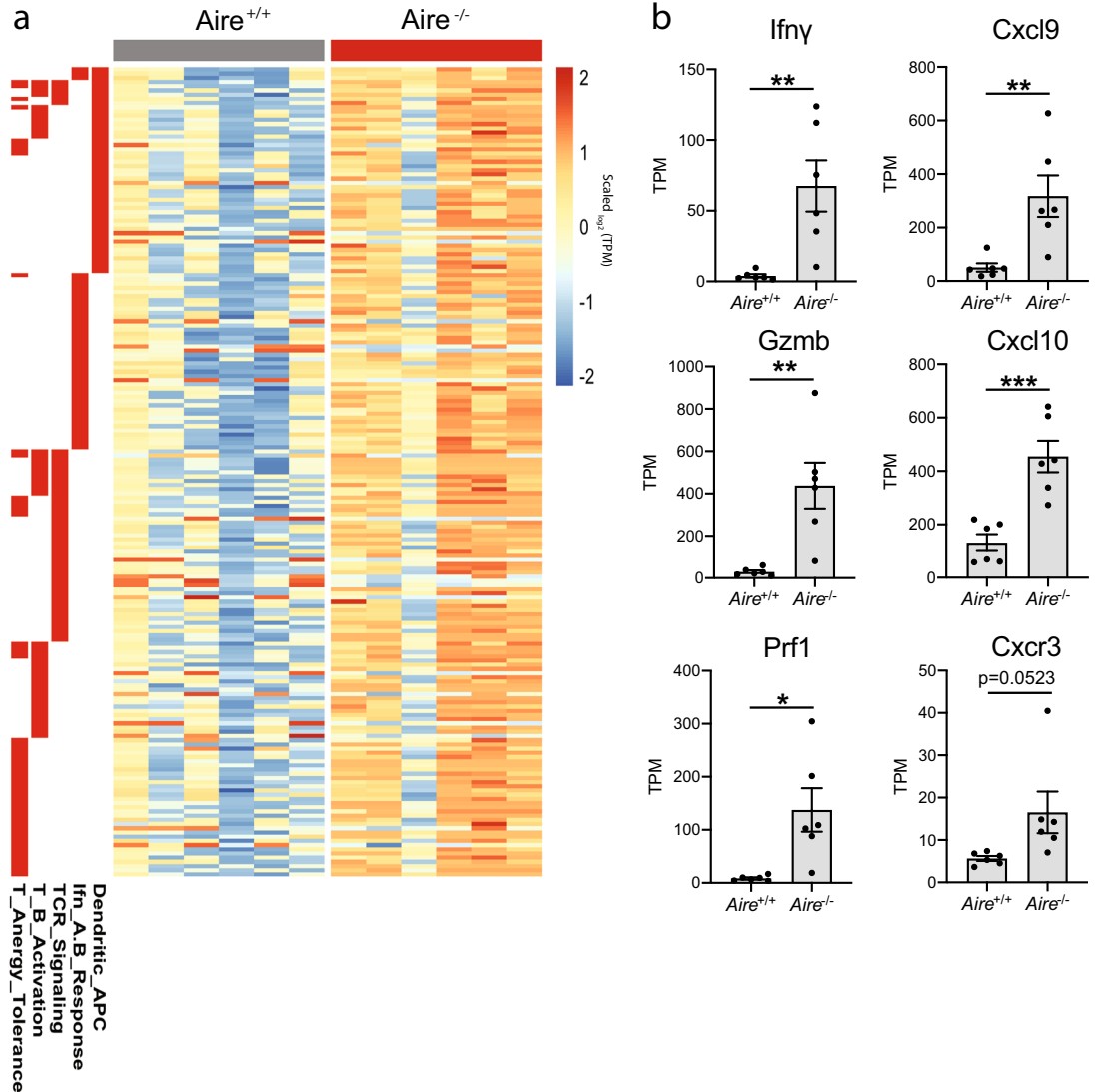

**Fig. 4 Tumors from _Aire_<sup>−/−</sup> mice treated with anti-CTLA4 have increased levels of cytotoxic genes. a** Heatmap depicting the differentially regulated genes in tumors from _Aire_<sup>+/+</sup> ($n = 6$) and _Aire_<sup>−/−</sup> ($n = 6$) mice treated with anti-CTLA4. Differentially regulated genes were defined with log2 fold changes of 1.5 in either up or down direction and with $p$ values < 0.01; TPM transcripts per million. The expression value of each gene was divided by the median expression of the same gene across all samples. The genes found to be differentially regulated were grouped according to different biological pathways (columns on left shaded in red). **b** Levels of Ifnγ, Gzmb, Prf1, Cxcl9, Cxcl10, and Cxcr3 in tumors from _Aire_<sup>+/+</sup> and _Aire_<sup>−/−</sup> mice. TPM, transcripts per million. Data are represented as mean ± SEM, (*$P < 0.05$; **$P < 0.01$; ***$P < 0.001$), by Student's $t$ test. TPM values are provided in Supplementary Data 2.

chains CDR3 repertoire diversity of the CD8<sup>+</sup> TILs and splenic CD8<sup>+</sup> T cells in knockout and wild-type mice using Shannon's diversity index. These results showed that CD8<sup>+</sup> TILs from _Aire_<sup>−/−</sup> mice treated with anti-PD1 or isotype have increased clonal diversity than the wild type (Fig. 7a). The diversity of the TCR repertoire of CD8<sup>+</sup> T cells in the tumor was significantly less than in the spleen (Supplementary Fig. 7a), probably as a result of T cell clones that were selectively expanded in the tumor. Interestingly, PD1 antagonism reduced the clonal diversity of CD8<sup>+</sup> TILs from _Aire_<sup>+/+</sup> and _Aire_<sup>−/−</sup> mice suggesting that selective clones are recruited to the tumor microenvironment to control tumor growth. Such decrease in clonal diversity has previously been observed in patients suffering from glioblastoma treated with anti-PD1 antibody and correlates with improved survival[40].

Given the increased CD8<sup>+</sup>TIL diversity observed in _Aire_<sup>−/−</sup> mice treated with anti-PD1, we next wanted to determine if the most expanded T cell clones were tumor reactive. To this end, we

used a JRT3 cell line, which normally does not express surface CD3 or the TCR α/β heterodimer, and introduced mCD28, mCD8α, and mCD8β. The cells were also engineered to express luciferase under an AP-1 promoter. To confirm that the cells would express luciferase upon activation, we treated the cells with PMA/Ionomycin and observed robust luciferase activity after 2 and 6 h (Supplementary Fig. 7b). To ensure that expression of a TCR α/β heterodimer would lead to activation of the AP-1 reporter, we engineered cells with a TCR that binds a known epitope on MC38 cells 41. To this end, we incubated the engineered cells with tumor cells which resulted in robust activation of the AP-1 promoter (~16 fold higher than a control TCR) as measured by luciferase activity (Supplementary Fig. 7c). We next introduced the TCR sequences of clones whose combined frequency accounted for more than 20% of the sequences analyzed (Supplementary Table 6) into a lentiviral expression vector and transduced the cells to evaluate their ability

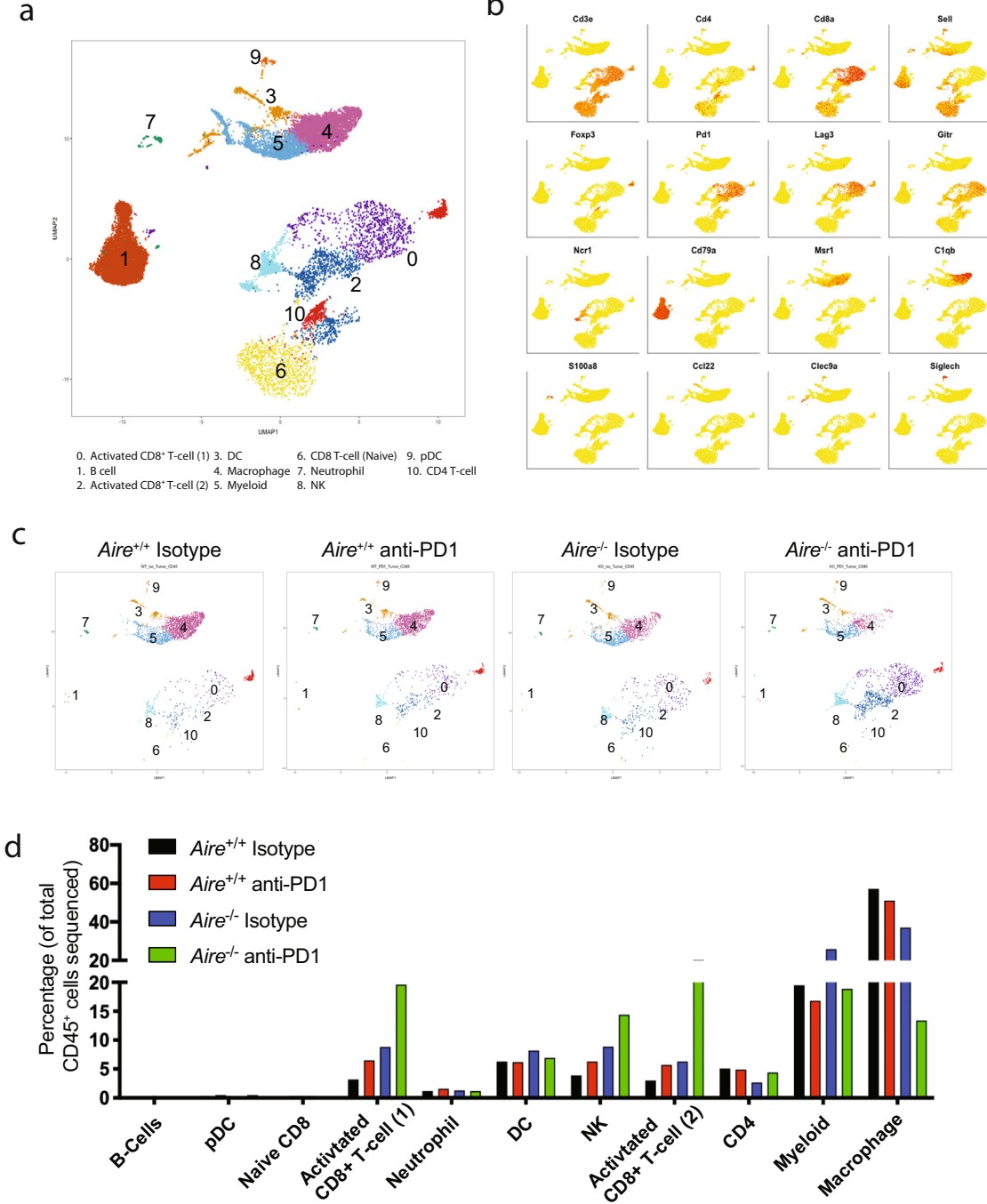

**Fig. 5 Single-cell analysis reveals changes in intra-tumoral cell populations in tumors from *Aire*$^{+/+}$ and *Aire*$^{-/-}$ mice. a** UMAP plot depicting the clusters from splenic and tumor CD45$^+$ cells. The cluster numbers correspond to individual cell populations described below the UMAP plot. **b** Individual UMAP plots from splenic and tumor CD45$^+$ cells depicting the expression of cell markers associated with cell clusters identified in Fig. 5a. T cells (CD3e$^+$), B cells (CD79a$^+$), macrophage/myeloid (Msr1$^+$ C1qb$^+$), dendritic cells (Clec9a$^+$), pDCs (Siglech$^+$), NK cells (Ncr1$^+$), and neutrophils (S100a8$^+$). **c** UMAP plots depicting the unsupervised clustering of tumor CD45$^+$ cells in *Aire*$^{+/+}$ and *Aire*$^{-/-}$ treated with Isotype or anti-PD1. **d** Bar graph depicting the composition of each cluster identified in tumor CD45$^+$ cells from *Aire*$^{+/+}$ and *Aire*$^{-/-}$ mice treated with Isotype or anti-PD1. Transcriptional changes in all cell types identified across all conditions are included in Supplementary Data 6.

to recognize tumors and engage downstream signaling pathways leading to induction of AP-1 luciferase activity. The most frequent clone, representing ~10% of the sequences analyzed, contained two TCR α sequences and a shared β sequence, and thus two independent cell lines, named TCR 1.1 and TCR 1.2 were generated to determine their tumor reactivity (Supplementary Table 6). We utilized two cell lines as negative controls, TCR

Control 1 and 2, expressing TCRs which were previously identified as non-tumor reactive. We observed surface expression of TCR α/β and CD3 in the cells transduced with the TCR-expressing vector, while the cells transduced with empty vector did not show any expression (Supplementary Fig. 7d). To test whether the engineered cells expressing the identified TCRs were indeed tumor reactive, we mixed them with tumor cells and

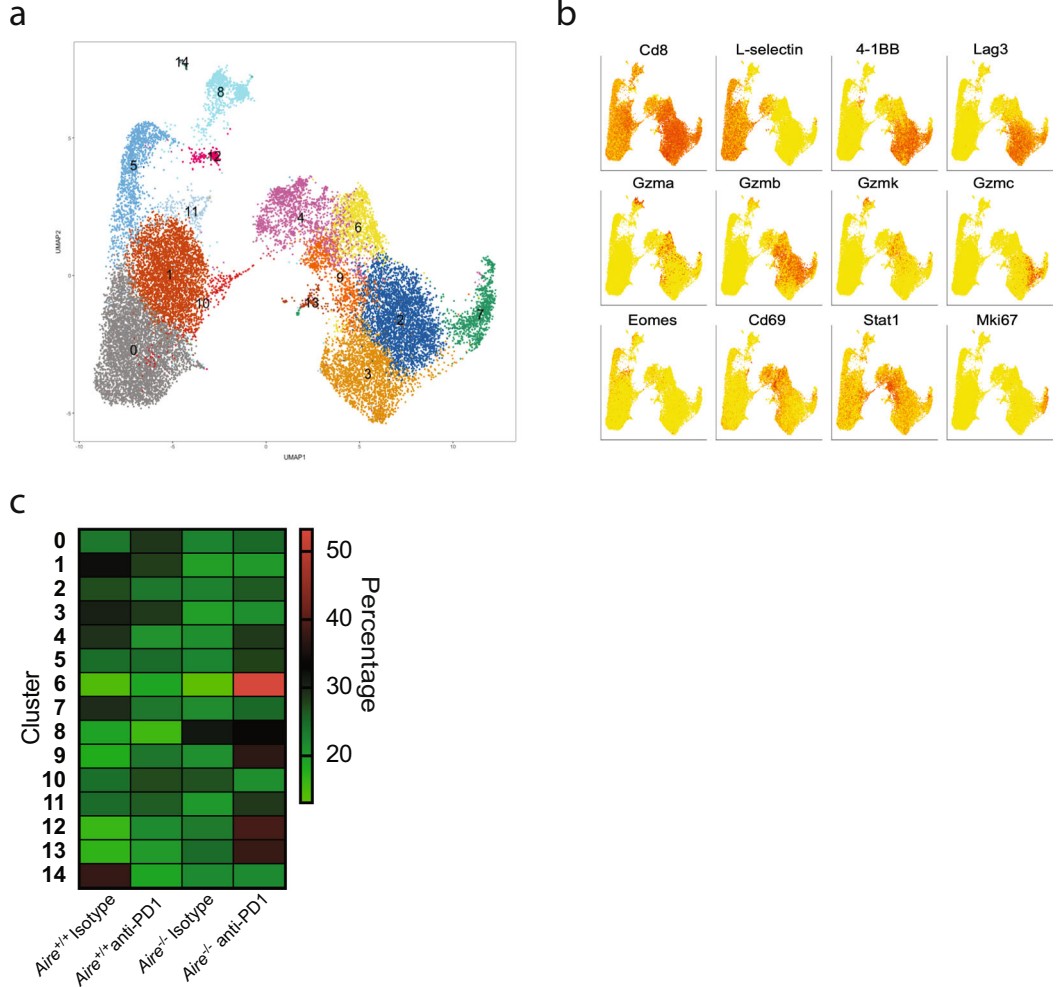

**Fig. 6 Sub-clustering of CD8$^+$ TILs show enrichment of highly cytotoxic cells in *Aire*$^{-/-}$ mice. a** UMAP plot of splenic and tumor CD8$^+$ T cells from *Aire*$^{+/+}$ and *Aire*$^{-/-}$ mice treated with Isotype or anti-PD1. Cluster numbers correspond to individual cell subpopulations. **b** Individual UMAP plots from splenic and tumor CD8$^+$ T cells depicting the expression of cell markers associated with cell clusters identified in Fig. 6a. **c** Heatmap depicting the composition of each cluster identified in the UMAP lot shown in Fig. 6a. Transcriptional changes for all clusters can be found in Supplementary Data 7.

assessed luciferase activity 6 h later. This line of experimentation showed that the engineered cell lines had significantly more luciferase activity in the presence of tumor cells compared with the negative controls (Fig. 7b). In addition, the fold changes in luciferase activity for the engineered cells over the non-tumor reactive TCR Control 1 cells ranged between ~12 and ~90.3-fold (Fig. 7b), comparable or higher to the previously identified TCR[41] (Supplementary Fig. 7c) highlighting the potency of the TCRs from the expanded clones in the tumors from *Aire*$^{-/-}$ mice. Interestingly, there was variation in the reactivity between the engineered cells expressing the identified TCRs. The cells expressing TCR 4 had the highest levels of reactivity compared with the cells expressing TCR 2 perhaps due to differences in the levels of antigen on the tumor cells. Altogether, these results suggest that PD1 blockade in *Aire*$^{-/-}$ results in the expansion of intra-tumoral CD8$^+$ T cells capable of recognizing tumor cells, potentially harboring unique tumor TCRs.

## Discussion

We report that deficiency in central tolerance in combination with immune-checkpoint blockade results in an enhanced immune response leading to significantly reduced tumor growth for both melanoma and colorectal cancers. Using the colorectal

tumor model MC38 and melanoma B16.F10, we showed that *Aire*$^{-/-}$ mice treated with anti-PD1 or anti-CTLA4 controlled tumor growth significantly better than the wild-type mice (Figs. 1b, 3a, b) and was accompanied by an increased infiltration of activated CD8$^+$ T cells into the tumors. Transcriptional profiling showed that tumors from *Aire*$^{-/-}$ had significantly more upregulated genes associated with antitumor activity such as Tnf, FasL, Xcl1, Cxcl9, Cxcl10, and Cxcr3 compared to wild-type mice (Fig. 2a–c and Supplementary Fig. 2a). In addition, anti-PD1 treatment resulted in the downregulation of many genes such as Ptp4a1 and Meis2 (Supplementary Fig. 2e) which have been implicated in the development and progression of cancer[28,29].

High dimensional analysis of tumor infiltrates from *Aire*$^{-/-}$ and *Aire*$^{+/+}$ mice treated with anti-PD1 showed dramatic changes in several key immune populations in addition to CD8$^+$ T cells, such as NK cells, and macrophages (Fig. 5c, d). The CD8$^+$ TILs from *Aire*$^{-/-}$ mice expressed high levels of the cytolytic molecules Gzma, Gzmb, and Perforin. Interestingly, we also observed higher levels of CD8$^+$ T cells expressing markers associated with tumor-resident memory cells such as Cd103 and Egr1. Although to date no reports have linked the generation of T$_{RM}$s to Aire deficiency, the presence of these cells has been shown to correlate with improved patient survival and several strategies are being explored to generate or expand this

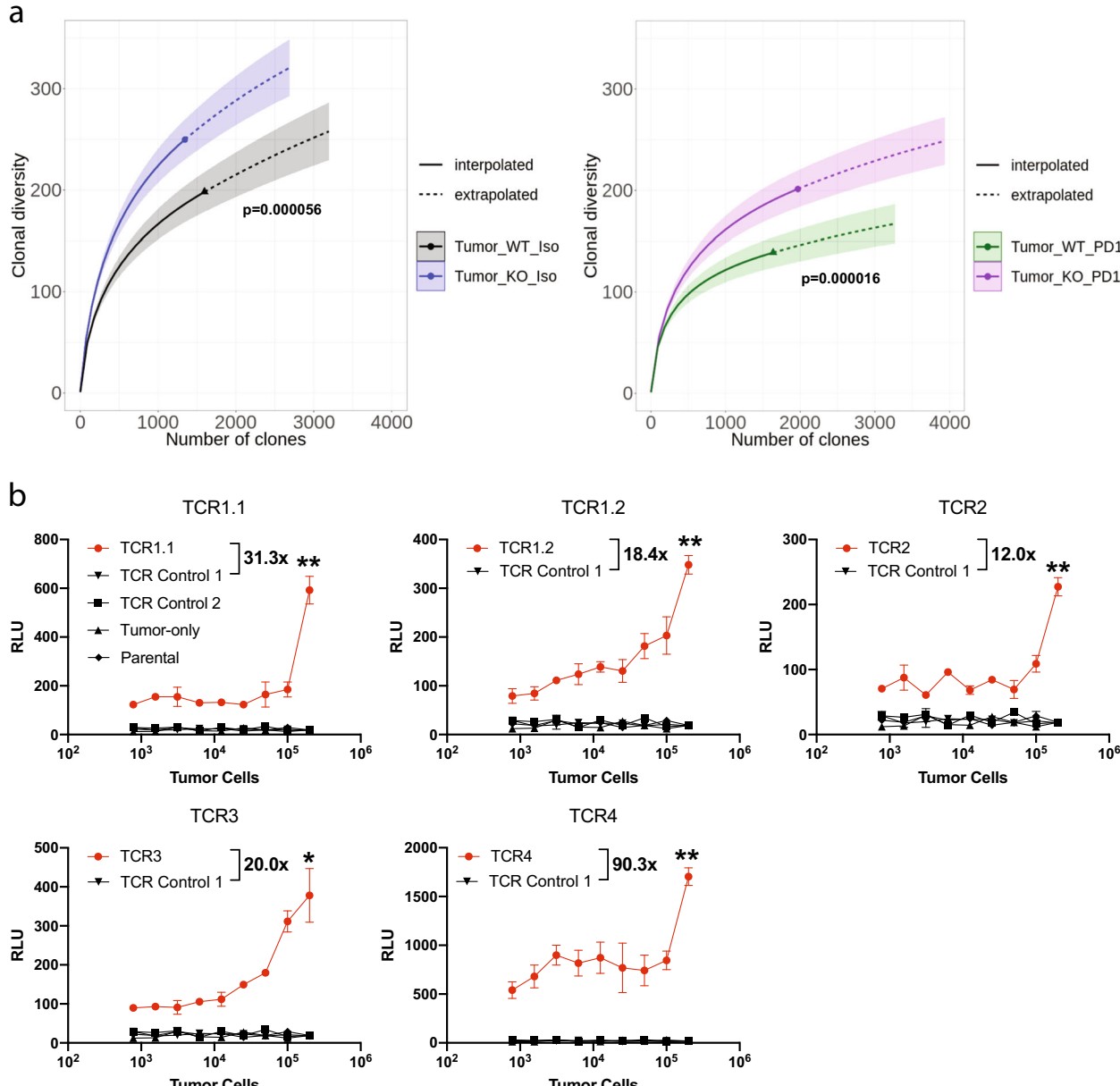

**Fig. 7 Increased expansion and clonal diversity of CD8⁺ TILs in *Aire⁻/⁻* mice. a** Shannon diversity was calculated for clones defined by the CDR3 sequences of TCR alpha and beta chains from *Aire⁺/⁺* and *Aire⁻/⁻* MC38 tumor bearing mice treated with isotype or anti-PD1 antibodies. Tumors were harvested on Day 19. Data were bootstrapped 50 times at the indicated sample sizes along the *x* axis. Interpolated lines (solid) were determined based on sample sizes smaller than the actual sample size. Extrapolated lines (dashed) were determined with sample sizes larger than the actual sample size. Extrapolated lines (dashed) were determined with sample sizes larger than the actual sample size. **b** Activation of the engineered T cells expressing the identified TCRs (Supplementary Table 6). T cells were mixed with increasing concentrations of tumor cells and luciferase activity was assessed 6 h later ($n = 3$ per group). Fold-change, indicated with × was determined by dividing the RLU values of each TCR by the values of TCR Control 1. Data are represented as mean ± SEM, (*$P < 0.05$; **$P < 0.01$; ***$P < 0.001$), by one-way ANOVA with Tukey's test. Full list of CDR3 sequences for all clones is included in Supplementary Data 8.

population[32,33,42]. In addition, we identified a population of CD8⁺ T cells that expressed high levels of type-I interferon stimulated genes (Fig. 6c and Supplementary Data 7). Several studies have shown that type-I IFNs act directly on CD8⁺ T cells to increase production of Ifnγ and promote their survival[39,43,44].

Given that the main role of AIRE in establishing central tolerance is to remove autoreactive CD4⁺ and CD8⁺ T cells, it was unexpected to find differences in the macrophage population of the tumors profiled. Tissue-associated macrophages represent a significant portion of the infiltrating leukocytes in the tumor microenvironment where they have been shown to play key

roles[45]. We observed that while tumors from *Aire⁻/⁻* mice treated with anti-PD1 had significantly less macrophages they expressed high levels of the M1-marker Nos2. Furthermore, these macrophages expressed higher levels of several type-I interferon stimulated genes which are characteristic of the M1 phenotype[46]. PD-1 blockade has recently been shown to promote the differentiation from M2 macrophages to M1 resulting in regression of lung metastases of osteosarcoma[47]. Furthermore, treatment with anti-PDL1 antibody resulted in the polarization of macrophages towards the pro-inflammatory phenotype[48]. Interestingly, autoreactive T cells promoted ocular infiltration of macrophages in a

Sjogren's syndrome model using $Aire^{-/-}$ mice[49]. It is conceivable, therefore, that macrophages in $Aire^{-/-}$ mice contain transcriptional programs which in combination with PD-1 blockade directs them toward the more tumoricidal M1 phenotype.

The transcriptional profiling of the tumor models used in this study revealed increased levels of the chemokines Cxcl9 and Cxcl10 in tumors from $Aire^{-/-}$ mice (Figs. 2b, 4b). Interestingly, we also observed increased levels of Cxcl10 in serum consistent with a previous study in humans suffering from APECED[27]. Cxcl9 and Cxcl10 have been shown to play significant roles in recruiting Cxcr3[+] T cells into the tumor and high levels correlate with improved patient survival[50]. A recent study showed that Cxcr3[+] CD8[+] T cells underwent rapid proliferation and were more activated in the presence of anti-PD1[26]. It is possible that the autoimmunity observed in $Aire^{-/-}$ mice and humans with APECED results in higher levels of circulating chemokines such as Cxcl9 and Cxcl10 produced by macrophages or dendritic cells which when present in tumors leads to enhanced recruitment of T cells, supporting an intricate feedback loop between innate and adaptive immune cells in orchestrating an enhanced antitumor response in $Aire^{-/-}$ mice.

In addition we showed that $Aire^{-/-}$ mice treated with anti-CTLA4 controlled melanoma growth better than wild type (Fig. 3a) as previously reported[16]. However, we observed increases in the percentages of both CD4[+] and CD8[+] T cells in $Aire^{-/-}$ treated with anti-CTLA4 over the wild type in contrast to what was previously observed. In depth transcriptional profiling of these tumors showed a higher number of upregulated antitumor genes in $Aire^{-/-}$ treated with anti-CTLA4 such as Ifng, Gzmb, and Perforin (Fig. 4a, b). Importantly, the observed synergy in tumor rejection extended to PD1 immunotherapy also, as evidenced by the enhanced tumor rejection in $Aire^{-/-}$ mice treated with anti-PD1 and increased cytolytic activity of CD8[+] TILs (Fig. 3b, c). The anti-melanoma response observed in $Aire^{-/-}$ mice has been attributed to the presence of autoreactive T cells, which would normally be eliminated by Aire-regulated central tolerance mechanisms. Aire regulates the expression of several self-antigens in mTECs expressed by melanoma cells such as gp100, TRP-1, and tyrosinase[17,18]. In addition, mTECs have been shown to express other tumor-associated antigens such as MUC1 and CEA[51]. It would be interesting to profile melanoma TILs in Aire-deficient mice to identify specific TCRs with enhanced antitumor activity. Our results showed a greater clonal diversity of CD8[+] TILs from $Aire^{-/-}$ mice (Fig. 7a) and more importantly the most highly expanded TILs were capable of recognizing tumor cells (Fig. 7b).

Taken together, the combination of immune-checkpoint blockade and breakdown in central tolerance orchestrate a sculpting of the tumor landscape that enhances tumor rejection and importantly provides an experimental platform for unmasking of potential novel cellular and molecular approaches. Although current inhibitors have greatly improved survival, their efficacy is limited to a minority of patients[52] and efforts to identify combination agents remains high. It was shown that depletion of mTECs using anti-RANKL resulted in enhanced killing of melanoma tumors and this has been proposed as a novel therapy[19,53]. Pursuing strategies that increase the number of antitumor T cells and modulate other cell types involved in tumor clearance will be of direct clinical relevance.

## Methods

**Mice**. 8-week-old $Aire^{+/+}$ and $Aire^{-/-}$ littermates (JAX, stock no. 004743) were purchased from Jackson Laboratories. All mice were used in accordance with guidelines from the Regeneron Pharmaceuticals Animal Care and Use Committee.

All experiments involving mice were performed with the approval of the Regeneron Pharmaceuticals Animal Care and Use Committee.

**Tumor models**. MC38 is a C57BL/6-derived colon tumor, and B16-10 is C57BL/6-derived melanoma tumor. The cells were maintained in complete DMEM composed of 10% FBS (Gibco, stock no. 26170043), and 1 mM penicillin/streptomycin (Gibco, Stock no. 15140148). Tumor cells were incubated as a monolayer in a humidified $CO_2$ incubator at 37 °C. Cells were maintained at ~70% confluency, and harvested with 0.05% trypsin (Gibco, Stock no. 15400054), washed and suspended in PBS before injection in mice. All cells lines used were mycoplasma free, and were authenticated using CellCheck (Short Tandem Repeat (STR) Profiling) by Idexx BioAnalytics.

**Tumor implantation, antibody treatment, and measurement**. Mice were injected with $4 \times 10^5$ MC38 cells, or $2 \times 10^5$ B16-F10 cells subcutaneous (s.c.) into the right flank. anti-PD1 blocking antibody (BioXcell, Clone RMP1-14, Stock no. BE0146) and isotype (BioXcell, Clone 2A3, Stock no. BE0089) were injected i.p. in 100 μL of PBS at a concentration of 5 mg/kg on days 0, 3, 7, 10, and 14. anti-CTLA4 blocking antibody (BioXcell, Clone 9D9, Stock no. BE0164) and isotype (BioXcell, Clone MPC-11, Stock no. BE0086) were injected i.p. in 100 μL of PBS at a concentration of 5 mg/kg on days 0, 3, 7, 10, and 14. Tumor volumes were measured blindly using a digital calipers and calculated using the formula $L \times W \times W \times 0.5$ where $L$ is the longest dimension and $W$ is the perpendicular dimension[54].

**Flow cytometry**. Tumors were enzymatically dissociated into single cells suspensions using Miltenyi Biotec's Mouse Tumor Dissociation Kit (Miltenyi, stock no. 130-096-730) and following manufacturer's instructions. Single-cell suspensions of spleen and lymph nodes were prepared by mechanical dispersion. Cell suspensions were incubated with LIVE/DEAD fixable blue dead cell stain (Invitrogen, Stock no. L23105) for 10 min at room temperature, washed twice with Cell Staining Buffer (Biolegend) and incubated with anti-CD16/CD32 blocking antibody (Biolegend, Stock no. 101302) for 10 min on ice. Cells were washed twice with Cell Staining Buffer, resuspended in staining antibody cocktail, and incubated on ice in the dark for 20 min. Cells were washed twice with Cell Staining Buffer and fixed overnight with Fixation Buffer (eBioscience Stock no. 00-8222-49). FoxP3-e450 (Clone FJK-16s, eBioscience, Cat. #48-5773-82), CD45-BV510 (Clone 30-F11, Biolegend, Cat. #103138), PD1-BV605 (Clone JA3, BD Bioscience, Cat. #563059), NK1.1-BV650 (Clone PKI36, BD Bioscience, Cat. #564143), GITR-BV711 (Clone DTA-1, BD Bioscience, Cat. #563390), CD4-BV786 (Clone RM4-5, BD Bioscience, Cat. #563727), CD11B-BUV395 (Clone M1/70, BD Bioscience, Cat. #563553), CD44-BUV737 (Clone IM7, BD Bioscience, Cat. #564392), CD8-BUV805 (Clone 53-6.7, BD Bioscience, Cat. #564920), CD3-Alexa 700 (Clone 17A2, eBioscience, Cat. #56-0032-82), Lag3-APC-Cy7 (Clone eBioC9B7W, eBioscience, Cat. #47-2231-82), Tim3-PE (Clone RMT3-23, eBioscience, Cat. #12-5870-82), CTLA4-PE-CF594 (Clone UC10-4F10-11, BD Bioscience, Cat. #564332), 2B4-PE-Cy7 (Clone m2B4 (B6)458.1, Biolegend, Cat. #133512), Live/Dead Blue-Fixable Blue Dye (Invivogen, Cat. #L-23105), and KLRG1-FITC (Clone 2F1, eBioscience, Cat. #11-5893-82).

**RNA preparation**. Tissue samples (tumors, spleens, and lymph nodes) were collected in RNAlater Stabilization Solution (ThermoFisher, Cat. #AM7020), and total RNA was purified from all samples using MagMAX-96 for Microarrays Total RNA Isolation Kit (Ambion by Life Technologies) according to manufacturer's specifications. Genomic DNA was removed using MagMAX™Turbo™DNase Buffer and TURBO DNase from the MagMAX kit listed above (Ambion by Life Technologies). Strand-specific mRNAseq libraries were prepared from total RNA using the KAPA Stranded mRNA-Seq Kits (Kapa Biosystems). Twelve-cycle PCR was performed to amplify libraries, and sequencing was performed on Illumina HiSeq®2500 (Illumina) by multiplexed single-read run with 80 cycles.

**RNAseq read mapping and statistical analysis of differentially expressed RNA**. Raw sequence data (BCL files) were converted to FASTQ format via Illumina bcl2fastq v2.17, and reads were decoded based on their barcodes and read quality was evaluated with FastQC (http://www.bioinformatics.babraham.ac.uk/projects/fastqc/). Reads were mapped to the mouse genome (mm10) using ArrayStudio software (OmicSoft, Cary, NC) allowing for two mismatches. Reads mapped to the exons of a gene were summed at the gene level. Differential expressed genes were identified by the DESeq2 R package[55] and significantly perturbed genes were defined with fold changes of at least 1.5 in either up or down direction and with $p$ values < 0.01.

**Single-cell RNA sequencing and read mapping**. $Aire^{+/+}$ and $Aire^{-/-}$ mice were implanted with $4 \times 10^5$ MC38 cells and treated with anti-PD1 blocking antibody and isotype at 5 mg/kg on days 0, 3, 7, 10, and 14. Tumors from 4 to 6 mice for each condition were collected on Day 19 post implant, processed into single-cell suspensions and pooled to obtain enough cells for single-cell RNA sequencing. Spleens were also collected from each condition, processed, and pooled. Single-cell suspensions of MC38 tumors or spleens were sorted for CD45[+] or CD8[+] and collected into tubes containing PBS with 0.04% BSA. The cell suspensions were

loaded on a Chromium Single Cell Instrument (10X Genomics) and RNA libraries were prepared using Chromium Single Cell 3′ Library, Gel Beads and Multiplex Kit (10X Genomics). Paired-end sequencing was performed on Illumina NextSeq500 where Read 1 was used for unique molecular identifier (UMI) and cell barcode while Read 2 was used for 55-bp transcript read. Sample demultiplexing, alignment, filtering, and UMI counting were performed on Cell Ranger Single-Cell Software Suite (10X Genomics).

**Single-cell data analysis**. Single-cell analysis was carried out using version 2 of the Seurat R package[56,57]. Cells with fewer than 200 genes detected or >20% of reads mapping to mitochondrial genes were discarded from analysis. Gene expression values for each cell were normalized and scaled due to variation in cell-cycle stage and mitochondrial rate, and were regressed out as described previously[56]. The number of UMI was also regressed out to correct for variation in sampling depth of these cells. The genes used for principal component analysis were the 1000 genes with the highest dispersion (variance to mean ratio) for genes with mean UMI between 0.0125 and 8 and variance above 0.5. Genes were divided into 20 bins of equal width based on their average expression and dispersion $z$ scores were calculated within these bins. Cells were then partitioned into clusters (Seurat FindClusters function) and visualized using the $t$-distributed stochastic neighbor embedding ($t$-SNE) algorithm (Seurat RunTSNE function) as described previously[56]. The first 15 principal components were used to run the $t$-SNE dimensionality reduction. The FindClusters function was run with a resolution parameter of 0.4, resulting in 14 clusters of cells. These clusters corresponded to naive $CD8^+$ T cells, B cells, activated $CD8^+$ T cells, macrophages, $CD8^+$ effector T cells, myeloid cells, dendritic cells (DCs), natural killer cells (NK), $CD4^+$ regulatory T cells, plasmacytoid dendritic cells (pDCs), and neutrophils. Cluster cell type identities were determined by examining marker genes specifically expressed more highly in each cluster (Seurat FindAllMarkers function) and expression of known immune marker genes (Seurat FeaturePlot function).

**5′ mouse TCR a/b cell partitioning, library preparation, sequence, and read alignment**. After sorting for $CD45^+$ or $CD8^+$ T cells from spleens or tumors, single cells were suspended in PBS with 0.04% BSA (KPL Immunoassays, Cat. # 50-61-00) were loaded on a Chromium Single Cell Instrument (10X Genomics). RNAseq and V(D)J libraries were prepared using Chromium Single Cell 5′ Libraries, Gel Beads and Multiplex Kit (10X Genomics). After amplification, cDNA was split into RNAseq and V(D)J library aliquots. To enrich the V(D)J library aliquot for TCR a/b, the cDNA was split into two 20 ng aliquots and amplified in two rounds using primers designed in-house. Specifically, for first round amplification the primers used were MP147 (ACACTCTTTCCCTACACGACGC) for short R1, MP122 (GGTGCTGTCCTGAGACCGAG) for mouse TRAC, and MP123 (CAATCTCTGCTTTTGATGGCTCAAAC) for mouse TRBC. For second round amplification, 20 ng aliquots from the first round were amplified using MP147 (ACACTCTTTCCCTACACGACGC) for short R1, MP130 (GTGACTGGAGTT CAGACGTGTGCTCTTCCGATCTTGGTACACAGCAGGTTCTGG) a nested R2 plus mouse TRAC, and MP131 (GTGACTGGAGTTCAGACGTGTGCTCT TCCGATCTGACCTTGGGTGGAGTCACATTTCTC) a nested R2 plus mouse TRABC. V(D)J libraries were prepared from 25 ng each mTRAC and mTRBC amplified cDNA. Paired-end sequencing was performed on Illumina NextSeq500 for RNAseq libraries (Read 1 26-bp for UMI and cell barcode, 8-bp i7 sample index, 0-bp i5, and Read 2 55-bp transcript read) and V(D)J libraries (Read 1 150-bp, 8-bp i7 sample index, 0-bp i5, and Read 2 150-bp read). For RNAseq libraries, Cell Ranger Single-Cell Software Suite (10X Genomics, v2.2.0) was used to perform sample demultiplexing, alignment, filtering, and UMI counting. The mouse mm10 genome assembly and RefSeq gene model for mouse were used for the alignment. For V(D)J libraries, Cell Ranger Single-Cell Software Suite (10X Genomics, v2.2.0) was used to perform sample demultiplexing, de novo assembly of read pairs into contigs, align and annotate contigs against all of the germline segment V(D)J reference sequences from IMGT, label and locate CDR3 regions, group clonotypes.

**Clonal diversity**. The diversity of the TCR repertoire (CDR3 TCR alpha and beta) was calculated using Shannon diversity index[58]. The TCR data for each condition tested are sampled 50 times at each sample size $n$ (ranging across the $x$ axis) giving the mean and standard deviation of the diversity at each given sample size. Interpolated means that $n$ is smaller than the actual observed sample size (the number of $CD8^+$T with full TCR sequences for each group). Extrapolation means $n$ is larger than the observed sample size with sampling including replacement to mimic a larger sample size.

**Pathway analysis**. KEGG pathways enriched in differentially expressed genes were determined using DAVID[59] with an FDR cutoff of 0.05. Heatmaps were generated for selected enriched pathways using log2 TPM expression values that were scaled across samples for each gene. A pseudo-count of 1 TPM added to all raw expression values to avoid infinite values.

**Cloning and expression of TCRs**. JRT3 cells were generated by transducing with lentiviruses expressing mouse CD8a, CD8b, and CD28. The cells were also transduced with a lentivirus expressing AP1.Luc (Qiagen, Cat. # 336851). The sequences obtained from the TCRseq of intra-tumoral $CD8^+$ T cells were cloned into lentivirus constructs and stable JRT3 cell lines were generated. To test for their ability to recognize tumor cells, the stable JRT3 cell lines were incubated with different amounts of MC38 cells for 6 h. Luciferase activity was measured by adding luciferase substrate (Promega Cat. E1500) and measuring luminescence on a luminometer.

**LDH assay**. Evaluation of $CD8^+$ and cytotoxic functional activity was performed using the Cytotoxicity Detection Kit LDH (Roche Applied Science, Basel, Switzerland) according to the manufacturer's protocol. Briefly, $CD8^+$ TILs were isolated from B16F10 tumor using a $CD8^+$ TIL isolation kit (Miltenyi Biotec, Cat. # 130-116-478). Isolation purity was assessed by flow cytometry prior to starting the LDH assay. Cytotoxicity was calculated according to the following formula: % cell lysis = (experimental value − effector spontaneous control − targets spontaneous control) × 100/(target maximum control − target spontaneous control), where "experimental" corresponds to the experimental signal value, "effector spontaneous control" to the spontaneous background signal value of the effector cells alone, "target spontaneous control" to the spontaneous background signal value of target tumor cells alone, and "target maximum control" to the maximum signal value of target cells in medium containing 1% Trixton X-100. Cytotoxicity assays were performed for varying effector cell (E) to target cell (T) ratios, and specific cytotoxicity lysis percentages are reported as the mean ± SEM of triplicate samples.

**Cytokine analysis**. Levels of cytokines in serum or tissue culture supernatant were measured using the V-PLEX Plus Mouse Cytokine 19-Plex Kit (MSD, Cat. # K15255G-1) according to the manufacturer's protocol.

**Statistics and reproducibility**. Significance values ($p$ values) were calculated with unpaired two-tailed Student's $t$ test for two-group comparisons or one-way ANOVA for multigroup comparisons with Tukey's comparison test. $P$ values of <0.05 were considered significant. $*P < 0.05$; $**P < 0.01$; $***P < 0.001$. Statistical analyses were performed with Graphpad Prism 8. Each experiment was repeated at least three times, and sample sizes and numbers are indicated in detail in each figure legend.

**Reporting summary**. Further information on research design is available in the Nature Research Reporting Summary linked to this article.

## Data availability

All bulk RNAseq and scRNAseq files have been deposited in the NCBI GEO Database and can be accessed using GSE151829, GSE151830, and GSE151831. Source data can be found in Supplementary Data 9.

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

## Acknowledgements

We are thankful to everyone who supported this work, in particular the Flow Cytometry Core at Regeneron Pharmaceuticals for their assistance in cell sorting, and to the Molecular Profiling Core for their help in sequencing the cells. We would also like to thank the Regeneron Pharmaceuticals Postdoctoral Committee for their invaluable input throughout this work.

## Author contributions

A.A.B. and S.H. designed the study. A.A.B. designed the experiments and performed all experiments with the help of S.K.A. and A.N. The bioinformatic analysis was performed by N.T.G. and W.Z. and G.S.A. provided advise for the analysis. A.J.M. and M.A.S. provided invaluable input into the study. A.A.B. and S.H. wrote the paper.

## Competing interests

This study was sponsored by Regeneron Pharmaceuticals, Inc. All authors are employees of Regeneron and may hold stock options in the company.
