## [Peer Review File · Communications Biology]

Reviewers' comments:

Reviewer #1 (Remarks to the Author):

This paper analyzes the impact of tumor growth/immune cell numbers and gene expression in AIRE knockout mice coupled with CTLA4 or PD-L1 blockade. Their results with PDL1 are in line with other publications, as referenced, and the CTLA4 data is as expected, but to my knowledge not published by other groups previously. The gene signatures they see are also expected and in line with other publications showing RNAseq data or protein data after checkpoint blockade treatment. The novelty of this paper lies in the construction of TCRs from AIREko mice, which may reveal novel antigenic targets and then the potential to make T cells for adoptive cell therapies, etc. This is something the authors allude to in the paper but do not state explicitly. If this is the rationale: how would you translate this to humans, as you can't make AIRE ko people? Unfortunately, figure 7 which contains this data is essentially uninterpretable. There is no description of panel A, no total n or any information, such that you could learn anything from the different boxes/columns. In the body of the manuscript, there is only a short description of fig 7A: "The distribution of the top 100 most frequent clones in each condition is shown in Figure 7A. Interestingly, we observed the most expansion of several unique clones in the tumors from Aire^{-/-} treated with anti-PD1. The first and second clone had a representation of ~12% and ~8% respectively." But that still does not tell us anything about the control groups to allow for comparisons. In fig 7B, tumor reactivity is shown of the re-engineered T cells. No idea which mice they came from. They are compared to some other re-engineered T cells, the origin of which is also unclear and may not even be from these experiments. It is not surprising you can isolate and make these TCR (albeit that it was a lot of work). What is the point? That the TCRs themselves are different among the groups? That was not shown. Just because you show anti-tumor effects in AIREko + checkpoint treated mice, it does not mean that the antigen-specificity is different. You could have the same antigen-specificity, but have low affinity clones in AIRE^{+/+} mice and checkpoint just increases frequency of high affinity clones in AIRE ko. This may also explain the difference in amount of tumor reactivity of the T cells shown (Y axes are all different). I can't interpret these data or determine if anything can be concluded from them because there is no description of the data. Since this is the only (potentially) novel aspect of the manuscript, this is not acceptable for publication.

Reviewer #2 (Remarks to the Author):

In this manuscript Benitez et al., report that breakdown of central tolerance (due to the deficiency of Aire) in combination with PD-1 blockade enhances antitumor immunity and consequently results in significantly reduced tumor growth. Specifically, the authors use two tumor models, colorectal MC38 and melanoma B16.F10 and compare their rejection in WT or Aire-deficient mice, which are either treated by PD-1 specific mAb or isotype control antibody. In some experiments the authors also use CTLA-4 blockade (figure 3).

Overall, this is a nice and important study, as it complements previous evidence demonstrating that Aire deficiency may be beneficial for cancer-immunotherapy. The study also demonstrates that the main effect of checkpoint inhibitors in both WT and Aire^{-/-} mice activation and recruitment of CD8⁺ T cells into the tumor, and that such an activation is significantly potentiated in Aire^{-/-} mice.

Furthermore, the authors also present interesting and rather comprehensive single-cell RNA-seq data of other immune infiltrates into the tumors, demonstrating that in addition to CD8⁺ T cells, there is also a decrease in macrophages in aPD-1 treated Aire^{-/-} mice, which are characteristic of the M1 "anti-tumor" phenotype.

The authors also identify an expanded population of tumor-specific TILs from aPD-1 treated Aire^{-/-} mice, which is evident to a lesser extent in mice treated with isotype control or WT treated with a-

PD1.

The main weakness of this study is that it is mostly descriptive with some of the claims being based on pure "correlation" rather than on fully worked out mechanism. This for instance includes:

- Demonstrating killing capacity of tumor cells by the isolated TILs cloned from the aPD1 Aire^{-/-} tumor
- Injecting CD8⁺ isolated TILs from aPD1 treated Aire^{-/-} mice to tumor bearing WT mice
- Injecting macrophages isolated from aPD1 treated Aire^{-/-} mice to tumor bearing WT mice

The authors also do not refer to the differences in Tregs between WT and Aire^{-/-} mice (isotype or aPD1 treated).

In addition, it seems that there's some inconsistency in the differences between Aire^{-/-} and WT mice in regards to T cells percentages without checkpoint inhibitors. In some cases, the difference between WT and Aire^{-/-} mice is significant (F3c), while in others it isn't – not really addressed by the authors, and somewhat confounding given that Aire^{-/-} mice are known to suffer from higher T cells infiltrates into tissues. This should be discussed and better presented in the paper.

Specific points:

- 1) Figure 1sA and 3sB is missing
- 2) Tumor model (MC38 or melanoma?) not specified for 10x experiments (Fig5/6)
- 3) Row 225-226: "We observed the expansion of several clones with more than three cells in all the conditions tested in tumors but not spleens" – data is missing/not shown
- 4) Row 258-259: "In addition, anti-PD1 treatment resulted in the downregulation of many genes such as Ptp4a1 and Meis2..." – data is missing/not shown
- 5) It is not clear why CTLA4 has been used only in the melanoma model in Fig 3 and not in the MC38 model in Fig1. For consistency, the authors should also use it for MC38 or explain why this is not possible to do
- 6) Gene names presented in Fig2A are not readable. The list of all differentially expressed genes can be available in a table in supplementary info, while Fig2A can highlight only the most interesting/relevant representatives
- 7) To make a stronger point, authors should incorporate protein validation, and better characterization by flow cytometry of the main populations involved in the tumor control (the increased CD8 and the decreased macrophages).
- 8) Please clarify difference in phenotype between activated CD8 and effector CD8.
- 9) In the single cell experiment, it would be valuable to comment on immunosuppressive populations like Treg, that look reduced in the Aire deficient vs the Aire wt mice by naked eye in figure 5c. It would be worth commenting, as previous evidence exists that there are Aire-dependent Treg specific for tumor antigens. (Malchow et al., Science, 2013)
- 10) What are the differences in the tumor models that would explain why PD-1 treatment did not enhance significantly the immune response in MC38 model (Fig1B) and yes in the B16.F10 melanoma? It would be valuable if authors would stain for PD-1 in TILs of these models, as well as other exhaustion markers.

Dear Dr. Karlsson Rosenthal,

We are delighted that our manuscript was peer-reviewed at Communications Biology and that the reviewers found the work of considerable potential interest. Likewise, we appreciate the feedback provided by the reviewers and we have edited the manuscript to address their questions raised and also included new data.

Reviewer 1:

As suggested by Reviewer #1, we elaborate further on the novelty of the paper as it pertains to TCR discovery in the *Aire*^{-/-} mice which may point to future studies to identify novel tumor-associated antigens. Figure 7 now includes a new analysis of the diversity (calculated using Shannon diversity index) in the tumor TCR repertoire between *Aire*^{+/+} and *Aire*^{-/-} mice treated with anti-PD1 or Isotype antibodies (Figure 7A and Figure S7A), supporting our original findings that *Aire*^{-/-} mice have a more diverse TCR repertoire than wild-type mice. In addition, we expanded the description of Figure 7 including additional information on the source of the TCRs we used to generate the cell lines as well the frequencies of each TCR and their sequences (Figure S7C).

Reviewer 2:

- a) Demonstrate whether isolated TILs from aPD1 *Aire*^{-/-} tumors have enhanced tumor killing
 - Per reviewer's suggestion, TILs were isolated from *Aire*^{+/+} or *Aire*^{-/-} mice implanted with B16.F10 tumors and treated with anti-PD1 or isotype antibodies. We then carried out a cytotoxicity assay by combining TILs with tumor cells and measuring the release of lactate dehydrogenase (LDH), which is released by damaged cells upon killing (Nguyen et al., 2016). This experiment showed that the TILs from *Aire*^{-/-} mice had enhanced tumor killing compared to *Aire*^{+/+} mice (Figure 3C). We also measured the levels of *Ifn* γ and *Tnf* α in the co-culture and showed that TILs from *Aire*^{-/-} had increased levels suggesting enhanced cytotoxic activity (Figure S3C), consistent with the increased anti-tumor response observed in these mice.
- b) Injecting CD8⁺ isolated TILs from aPD1 treated *Aire*^{-/-} mice to tumor bearing WT mice
 - From previous experiments we have carried out, the total number of TILs isolated is very small and not enough to transfer to other animals. However, we were able to isolate enough TILs to perform cytotoxic assays as previously described in prior point.
- c) Injecting macrophages isolated from aPD1 treated *Aire*^{-/-} mice to tumor bearing WT mice
 - From prior studies, we have not been able to isolate and transfer sufficient numbers of infiltrated macrophages from tumors and moreover it has been shown that the isolation of tumor-associated macrophages (TAMs) and subsequent manipulation drastically change the transcriptome and polarization status of these cells (Orecchioni et al., 2019, Sica and Mantovani, 2012). To this end, for technical and biological reasons we're not able to perform these questions with current tools.
- d) The authors also do not refer to the differences in Tregs between WT and *Aire*^{-/-} mice (isotype or aPD1 treated).

- We have now included new figures to demonstrate the increase in the ratio of CD8 to Tregs in tumors from *Aire*^{-/-} treated with anti-PD1 or anti-CTLA4 (Figures S1C and S3C).
- e) In addition, it seems that there's some inconsistency in the differences between *Aire*^{-/-} and WT mice in regard to T cells percentages without checkpoint inhibitors. In some cases, the difference between WT and *Aire*^{-/-} mice is significant (F3c), while in others it isn't – not really addressed by the authors, and somewhat confounding given that *Aire*^{-/-} mice are known to suffer from higher T cells infiltrates into tissues. This should be discussed and better presented in the paper.
- As the reviewer points, *Aire*^{-/-} mice have increased infiltration into many different tissues, albeit in different degrees, such as the salivary and lacrimal glands, reproductive organs, and others (Hubert et al., 2009, Anderson et al., 2002, Kuroda et al., 2005). Within the tumor environment we find that the addition of immune-checkpoint blockade greatly enhances the recruitment of CD8+ T cells in *Aire*^{-/-} mice. The differences noted by the reviewer between *Aire*^{+/+} and *Aire*^{-/-} mice without check-point inhibitors likely reflect the differences between tumor models and source of original tissue (MC38; colon adenocarcinoma) and B16.F10 (melanoma) used, which speaks to the antigenic-TCR driven response of the model.

Specific points raised by Reviewer 2:

- 1) Figure 1SA and 3SB is missing.
 - These figures were not properly transferred when the PDF file was generated. Figure 1SA is now included in the PDF file while figure 3SB was removed to incorporate new data.
- 2) Tumor model (MC38 or melanoma?) not specified for 10x experiments (Fig5/6).
 - We used the MC38 tumor model for these experiments. This has now been addressed in the main text, figure legend, and materials and methods.
- 3) Row 225-226: “We observed the expansion of several clones with more than three cells in all the conditions tested in tumors but not spleens” – data is missing/not shown
 - We performed Shannon diversity index to better understand the differences in the TCR repertoire between *Aire*^{+/+} and *Aire*^{-/-} mice treated with isotype or anti-PD1 antibodies (Figure 7A and Figure S7A). These results showed that *Aire*^{-/-} mice have a more diverse TCR repertoire than wild-type mice.
- 4) Row 258-259: “In addition, anti-PD1 treatment resulted in the downregulation of many genes such as *Ptp4a1* and *Meis2*...” – data is missing/not shown
 - Bar graphs for these two genes have been added to Figure S2E.
- 5) It is not clear why CTLA4 has been used only in the melanoma model in Fig 3 and not in the MC38 model in Fig1. For consistency, the authors should also use it for MC38 or explain why this is not possible to do.
 - This experiment showed reduced tumor growth in *Aire*^{-/-} mice treated with anti-CTLA4 as compared to wild-type mice and it's included in Figure S3D.
- 6) Gene names presented in Fig2A are not readable. The list of all differentially expressed genes can be available in a table in supplementary info, while Fig2A can highlight only the most interesting/relevant representatives

- The genes in the original Figure 2A will be provided in a table in supplementary info as the reviewer suggested. The original Figure 2A was moved to supplementary Figure S2A to highlight all the genes that were differentially regulated. The current Figure 2A and 2B highlight a subset of differentially regulated genes.
- 7) To make a stronger point, authors should incorporate protein validation, and better characterization by flow cytometry of the main populations involved in the tumor control (the increased CD8 and the decreased macrophages).
- As suggested by the reviewer we added protein validation for the decrease in macrophages (Figure S5H). We also observed the increase in the CD8⁺ population described in Figure 1C.
- 8) Please clarify difference in phenotype between activated CD8 and effector CD8.
- The clustering done in Figure 5 for the single-cell RNAseq analysis of tumor infiltrates was done by identifying the highest expressed genes that defined each population. Below is a figure that shows the genes that were the most highly expressed in each population. These two populations differ in their expression of select genes, such as Pdc1 (Pd-1) and Lag3, but also share a significant part of their transcriptional profile as they represent two clusters of CD8 T cells with different activation status – to this end, we've renamed them Activated CD8 (1) and Activated CD8 (2).

- 9) In the single cell experiment, it would be valuable to comment on immunosuppressive populations like Treg, that look reduced in the Aire deficient vs the Aire wt mice by naked eye in figure 5c. It would be worth commenting, as previous evidence exists that there are Aire-dependent Treg specific for tumor antigens. (Malchow et al., Science, 2013)
- Indeed Malchow et al. described an endogenous Aire-dependent Treg population (called MJ23) that was enriched in mice with prostate cancer, a tumor model different from the ones tested here. We examined the TCR profile of Tregs in our single-cell RNAseq experiment and were not able to detect the TCR sequence described for the MJ23 population. One possibility for this is that the MJ23 population is not present in very high abundance to be detected by the scRNAseq methodology used. Perhaps doing scRNAseq exclusively on tumor Tregs would identify such population. Another possibility is that Malchow et al. used a prostate tumor model instead of the colorectal or melanoma model used in our studies. However, it is certainly possible that some of the Tregs we identified are Aire-dependent and could be coopted by the tumor models used.
- 10) What are the differences in the tumor models that would explain why PD-1 treatment did not enhance significantly the immune response in MC38 model (Fig1B) and yes in the B16.F10

melanoma? It would be valuable if authors would stain for PD-1 in TILs of these models, as well as other exhaustion markers.

- The effect of anti-PD1 immunotherapy varies with different tumor models. Other groups have shown that B16.F10 tumor respond slightly better to PD1 blockade than MC38 tumors (Chen et al., 2015). Other reports have demonstrated that different subsets of immune cells are recruited by different tumors and shape the responsiveness to immunotherapy (Tang and Zheng, 2018).

References

- ANDERSON, M. S., VENANZI, E. S., KLEIN, L., CHEN, Z., BERZINS, S. P., TURLEY, S. J., VON BOEHMER, H., BRONSON, R., DIERICH, A., BENOIST, C. & MATHIS, D. 2002. Projection of an immunological self shadow within the thymus by the aire protein. *Science*, 298, 1395-401.
- CHEN, S., LEE, L. F., FISHER, T. S., JESSEN, B., ELLIOTT, M., EVERING, W., LOGRONIO, K., TU, G. H., TSAPARIKOS, K., LI, X., WANG, H., YING, C., XIONG, M., VANARSDALE, T. & LIN, J. C. 2015. Combination of 4-1BB agonist and PD-1 antagonist promotes antitumor effector/memory CD8 T cells in a poorly immunogenic tumor model. *Cancer Immunol Res*, 3, 149-60.
- HUBERT, F. X., KINKEL, S. A., CREWETHER, P. E., CANNON, P. Z., WEBSTER, K. E., LINK, M., UIBO, R., O'BRYAN, M. K., MEAGER, A., FOREHAN, S. P., SMYTH, G. K., MITTAZ, L., ANTONARAKIS, S. E., PETERSON, P., HEATH, W. R. & SCOTT, H. S. 2009. Aire-deficient C57BL/6 mice mimicking the common human 13-base pair deletion mutation present with only a mild autoimmune phenotype. *J Immunol*, 182, 3902-18.
- KURODA, N., MITANI, T., TAKEDA, N., ISHIMARU, N., ARAKAKI, R., HAYASHI, Y., BANDO, Y., IZUMI, K., TAKAHASHI, T., NOMURA, T., SAKAGUCHI, S., UENO, T., TAKAHAMA, Y., UCHIDA, D., SUN, S., KAJIURA, F., MOURI, Y., HAN, H., MATSUSHIMA, A., YAMADA, G. & MATSUMOTO, M. 2005. Development of autoimmunity against transcriptionally unrepressed target antigen in the thymus of Aire-deficient mice. *J Immunol*, 174, 1862-70.
- NGUYEN, H. H., KIM, T., SONG, S. Y., PARK, S., CHO, H. H., JUNG, S. H., AHN, J. S., KIM, H. J., LEE, J. J., KIM, H. O., CHO, J. H. & YANG, D. H. 2016. Naive CD8(+) T cell derived tumor-specific cytotoxic effectors as a potential remedy for overcoming TGF-beta immunosuppression in the tumor microenvironment. *Sci Rep*, 6, 28208.
- ORECCHIONI, M., GHOSHEH, Y., PRAMOD, A. B. & LEY, K. 2019. Macrophage Polarization: Different Gene Signatures in M1(LPS+) vs. Classically and M2(LPS-) vs. Alternatively Activated Macrophages. *Front Immunol*, 10, 1084.
- SICA, A. & MANTOVANI, A. 2012. Macrophage plasticity and polarization: in vivo veritas. *J Clin Invest*, 122, 787-95.
- TANG, F. & ZHENG, P. 2018. Tumor cells versus host immune cells: whose PD-L1 contributes to PD-1/PD-L1 blockade mediated cancer immunotherapy? *Cell Biosci*, 8, 34.

Reviewers' comments:

Reviewer #1 (Remarks to the Author):

In this revised manuscript, the authors have added data to better show TCR diversity in AIRE KO mice vs. WT mice, Fig 7. The legend for the new figure 7 says that "CD3ab diversity" was analyzed. There is no such thing as CD3 alpha beta. What was analyzed here? At what timepoint was the tumor taken down?

There is very little description as to how many cells were analyzed, from how many animals, etc. Lines 218-220: "sequenced intratumoral or splenic CD8+ T cells from Aire+/+ and Aire-/- treated with anti-PD1 or isotype resulting in ~9,000 and ~10,000 individually sequenced cells from each tissue respectively." There are 4 tissues here, spleen and tumor from both KO and WT animals. How many cells and from how many animals were sequenced? If so many single cells were sequenced, why are only ~2000 clones presented in the non-extrapolated data, figs 7, 7S?

The luciferase assay is not common for showing T cell activity. There is no positive control in these assays to show what amount of RLU is to be expected with known TCR stimulation. In fig S7, PMA treated cells have a RLU of 70,000, whereas the engineered T cells that may be reacting to antigen are being scored as positive when RLU is between 200-1700. That is 350x-40x less than the PMA data, so hard to know if those are really positive.

There is no legend for Fig S7.

Figure 2 shows data from tumors in AIRE ko and WT mice. Between Fig 2 and 2S, the only data from animals treated with anti-PD1 is 2SD. Because, AIRE ko and WT tumors grow at the same rate as shown in Fig 1B, why are there so many differences in these tumors in fig 2 for both the amount of T cells and function of T cells, esp as the number of T cells shown by flow in fig 8 is the same, e.g. compare 2C with 1C.

Reviewer #2 (Remarks to the Author):

The authors have sufficiently addressed critical the points that were raised previously, therefore I find the study acceptable for publication in CommsBio

Dear Dr. Karlsson Rosenthal,

We are delighted that our manuscript was peer-reviewed at Communications Biology and that the reviewers found the work of considerable potential interest. Likewise, we appreciate the feedback provided by the reviewers and we have edited the manuscript to address their questions raised and also included new data.

Question #1. In this revised manuscript, the authors have added data to better show TCR diversity in AIRE KO mice vs. WT mice, Fig 7. The legend for the new figure 7 says that “CD3ab diversity” was analyzed. There is no such thing as CD3 alpha beta. What was analyzed here? At what timepoint was the tumor taken down?

We thank the reviewer for raising this concern. We have updated the figure legend to more specifically define the type of analysis performed. Below please find the original legend and the updated legend.

Original

Figure 7. Increased expansion of CD8⁺ TILs in *Aire*^{-/-} mice. (A) Shannon’s entropy was applied to determine the CD3αβ diversity in MC38 tumors from *Aire*^{+/+} and *Aire*^{-/-} mice treated with isotype or anti-PD1 antibodies. Data were sampled 50 times at the indicated sample sizes along the x-axis. Interpolated lines (solid) were determined based on sample sizes smaller than the actual sample size. Extrapolated lines (dashed) were determined with sample sizes larger than the actual sample size. (B) Graphs depicting the activation of the engineered T cells expressing the identified TCRs. T-cells were mixed with increasing concentrations of tumor cells and luciferase activity was assessed 6 hours later. Data are represented as mean ± SEM, (*, P < 0.05; **, P < 0.01; ***, P < 0.001).

Changed

Figure 7. Increased expansion of CD8⁺ TILs in *Aire*^{-/-} mice. (A) Shannon diversity was calculated for clones defined by the CDR3 sequences of TCR alpha and beta chains from MC38 tumors from *Aire*^{+/+} and *Aire*^{-/-} mice treated with isotype or anti-PD1 antibodies on Day 19. Data were bootstrapped 50 times at the indicated sample sizes along the x-axis. Interpolated lines (solid) were determined based on sample sizes smaller than the actual sample size. Extrapolated lines (dashed) were determined with sample sizes larger than the actual sample size. Extrapolated lines (dashed) were determined with sample sizes larger than the actual sample size. (B) Graphs depicting the activation of the engineered T cells expressing the identified TCRs. T-cells were mixed with increasing concentrations of tumor cells and luciferase activity was assessed 6 hours later. Data are represented as mean ± SEM, (*, P < 0.05; **, P < 0.01; ***, P < 0.001).

Question #2. There is very little description as to how many cells were analyzed, from how many animals, etc. Lines 218-220: “sequenced intratumoral or splenic CD8⁺ T cells from *Aire*^{+/+} and *Aire*^{-/-} treated with anti-PD1 or isotype resulting in ~9, 000 and ~10, 000 individually sequenced cells from each tissue respectively.” There are 4 tissues here,

spleen and tumor from both KO and WT animals. How many cells and from how many animals were sequenced? If so many single cells were sequenced, why are only ~2000 clones presented in the non-extrapolated data, figs 7, 7S?

The table below lists the number of cells that were sequenced from the CD45⁺ sort. The Raw Cells refers to the exact of number of cells captured and sequenced, and QC Pass refers to the number of cells that passed the QC criteria described in the Materials & Methods section under Single-Cell Data Analysis. This table has been included in Figure S5B.

	Tissue	Sorted	Treatment	Raw Cells	QC Pass
Aire ^{+/+}	Tumor	CD45	Isotype	2936	2930
		CD45	PD1	2211	2208
	Spleen	CD45	Isotype	2119	2095
		CD45	PD1	2117	2088
Aire ^{-/-}	Tumor	CD45	Isotype	1860	1855
		CD45	PD1	2076	2075
	Spleen	CD45	Isotype	2048	2023
		CD45	PD1	2402	2393

The table below lists the number of cells that were sequenced from the CD8⁺ sort. In this table, the last column lists the number of cells from which complete CDR3 sequences of TCR alpha and beta chains were obtained. This table has been included in Figure S6A.

The number of cells is represented in the x-axis of Figures 7A, and S7A.

	Tissue	Sorted	Treatment	Raw Cells	QC Pass	Cells with TCRA and TCRB sequences
Aire ^{+/+}	Tumor	CD8	Isotype	2688	2684	1596
		CD8	PD1	2222	2221	1637
	Spleen	CD8	Isotype	2902	2898	2007
		CD8	PD1	2966	2944	2081
Aire ^{-/-}	Tumor	CD8	Isotype	1860	1858	1345
		CD8	PD1	2214	2212	1968
	Spleen	CD8	Isotype	2355	2339	1715

		CD8	PD1	2633	2629	1901
--	--	-----	-----	------	------	------

In addition, we updated the Materials & Methods section to specify the number of animals used in this study.

Question #3. The luciferase assay is not common for showing T cell activity. There is no positive control in these assays to show what amount of RLU is to be expected with known TCR stimulation. In fig S7, PMA treated cells have a RLU of 70,000, whereas the engineered T cells that may be reacting to antigen are being scored as positive when RLU is between 200-1700. That is 350x-40x less than the PMA data, so hard to know if those are really positive.

We used PMA to ensure that the luciferase reporter construct functioned as intended when the engineered cells were stimulated. In other experiments, we have generated cells expressing TCRs that recognize known peptides derived from tumors or viruses as positive controls with different RLU values. This difference in RLU values can be attributed to the abundance of peptide presented by the antigen presenting cell (Corse et al., 2011) or the sensitivity of TCRs to different epitopes.

Question #4. There is no legend for Fig S7.

We thank the reviewer for pointing out this figure legend missing. This figure legend did not properly transfer onto the PDF file when the conversion was done. Below please find the figure legend for Figure S7.

Figure 7S. Related to Figure 7. (A) Shannon diversity was calculated for clones defined by the CDR3 sequences of TCR alpha and beta chains from spleens from *Aire*^{+/+} and *Aire*^{-/-} MC38-tumor bearing mice treated with isotype or anti-PD1 antibodies on Day 19. Data were bootstrapped 50 times at the indicated sample sizes along the x-axis. Interpolated lines (solid) were determined based on sample sizes smaller than the actual sample size. Extrapolated lines (dashed) were determined with sample sizes larger than the actual sample size. (B) Activation of JRT3 cells with PMA leads to production of Luciferase after 2 and 6 hours. (C) Sequences of TCRs isolated from tumors from *Aire*^{-/-} treated with anti-PD1. Size represents the number of times the sequence was identified for each TCR and percentage represents the frequency of each TCR out of the total number of TCRs identified. (D) Representative FACS plots of JRT3 cells expressing the cloned TCR. Cells were transduced with lentiviruses expressing either a TCR or none as a control. Cells were stained for CD8 α , CD8 β , CD28, CD3, and TCR α/β and expression was confirmed using flow-cytometry.

Question #5. Figure 2 shows data from tumors in AIRE ko and WT mice. Between Fig 2 and 2S, the only data from animals treated with anti-PD1 is 2SD. Because, AIRE ko and WT tumors grow at the same rate as shown in Fig 1B, why are there so many

differences in these tumors in fig 2 for both the amount of T cells and function of T cells, esp as the number of T cells shown by flow in fig 8 is the same, e.g. compare 2C with 1C.

We thank the reviewer for pointing out this concern. The data shown in Figure 2 is from tumors from *Aire*^{+/+} and *Aire*^{-/-} mice both treated with anti-PD1. We have now corrected this in the Figure 2 legend (highlighted in green), and in the manuscript to describe the experiment in more detail.

(Lines 103-104). “To further profile the observed antitumor response, we performed RNAseq in tumors from *Aire*^{+/+} and *Aire*^{-/-} mice treated with anti-PD1”.

References

CORSE, E., GOTTSCHALK, R. A. & ALLISON, J. P. 2011. Strength of TCR-peptide/MHC interactions and in vivo T cell responses. *J Immunol*, 186, 5039-45.